# Subsurface A-site vacancy activates lattice oxygen in perovskite ferrites for methane anaerobic oxidation to syngas

Jiahui He[1,2,7], Tengjiao Wang[3,7], Xueqian Bi[1,4], Yubo Tian[1,5], Chuande Huang [1]✉, Weibin Xu[1,6], Yue Hu[1,6], Zhen Wang[1,6], Bo Jiang [3]✉, Yuming Gao[3], Yanyan Zhu [2]✉ & Xiaodong Wang [1]✉

Tuning the oxygen activity in perovskite oxides ($ABO_3$) is promising to surmount the trade-off between activity and selectivity in redox reactions. However, this remains challenging due to the limited understanding in its activation mechanism. Herein, we propose the discovery that generating subsurface A-site cation ($La_{sub.}$) vacancy beneath surface Fe-O layer greatly improved the oxygen activity in $LaFeO_3$, rendering enhanced methane conversion that is 2.9-fold higher than stoichiometric $LaFeO_3$ while maintaining high syngas selectivity of 98% in anaerobic oxidation. Experimental and theoretical studies reveal that absence of $La_{sub.}$-O interaction lowered the electron density over oxygen and improved the oxygen mobility, which reduced the barrier for C-H bond cleavage and promoted the oxidation of C-atom, substantially boosting methane-to-syngas conversion. This discovery highlights the importance of A-site cations in modulating electronic state of oxygen, which is fundamentally different from the traditional scheme that mainly credits the redox activity to B-site cations and can pave a new avenue for designing prospective redox catalysts.

Tuning the oxygen activity in metal oxides offers opportunities to manufacture prospective catalysts for reactions proceeding via the Mars–van Krevelen mechanism[1–3]. However, it was usually found that the oxides with high oxidizing capability display low selectivity, and vice versa, which poses huge obstacle for designing an efficient redox catalyst[4,5]. The perovskite oxides (structural formula: $ABO_3$) has attracted particular attention in redox cycling reactions due to their capacity to accommodate different cations in single perovskite matrix[6–8]. The oxygen activity in $ABO_3$ can be widely adjusted by selecting suitable A- and B-site cations, substantially endows these

oxides outstanding redox properties and superior performance in various processes, e.g., selective methane oxidation[9,10], dehydrogenation of light alkanes[11,12], oxygen evolution reaction[13–15], and CO oxidation[16,17]. Therefore, disclosing the underlying mechanism affecting the oxygen activity in perovskite oxides would be promising to surmount the trade-off between activity and selectivity during redox reactions.

The current understanding mainly credits the outstanding oxygen activity in perovskites to transition metals at the B-site, since the A-site is generally occupied by redox-inert alkaline, alkaline-earth, or

[1]CAS Key Laboratory of Science and Technology on Applied Catalysis, Dalian Institute of Chemical Physics, Chinese Academy of Sciences, Dalian 116023, China. [2]School of Chemical Engineering, Northwest University, International Scientific and Technological Cooperation Base of MOST for Clean Utilization of Hydrocarbon Resources, Chemical Engineering Research Center for the Ministry of Education for Advance Use Technology of Shanbei Energy, Xi'an 710069, China. [3]Key Laboratory of Ocean Energy Utilization and Energy Conservation of Ministry of Education, Dalian University of Technology, Dalian 116023, China. [4]College of Environmental Science and Engineering, Dalian Maritime University, Dalian 116026, China. [5]School of Chemical Engineering, Zhengzhou University, Zhengzhou 450001, P. R. China. [6]School of Chemical Engineering, University of Chinese Academy of Science, Beijing 100049, China. [7]These authors contributed equally: Jiahui He, Tengjiao Wang. ✉e-mail: huangchuande@dicp.ac.cn; bjiang@dlut.edu.cn; zhuyanyan@nwu.edu.cn; xdwang@dicp.ac.cn

lanthanide cations[18–20]. To this end, great effort, such as constructing asymmetric $B_1$-O-$B_2$ interaction, acid etching, and increase B/A ratio in perovskite oxides, has been taken to optimize the redox activity by regulating B-O interaction or terminating catalyst surface by more B-site cations[21–23]. Compared to the dazzling glory of B-site cations, the functionalization of A-site cations is normally explained by an indirect manner of generating oxygen vacancies, adjusting the valence state of B-site cations, or changing the crystal structure[24–26]. However, from the view of structure analysis (Fig. 1a), a lattice oxygen in the bulk ($O_{bulk}$), taken cubic structure as an example, is coordinated to four A-site cations and two B-site cations, which provides possibilities for A-site cations to directly regulate the electronic state of oxygen. This hypothesis is validated by some recent observations. Gong et al.[27] found that the substitution of $La^{3+}$ by $Ce^{3+}$ in $La_xCe_{1-x}FeO_3$ showed a pronounced effect on oxygen mobility without changing the concentration of oxygen vacancies, Fe valence state, and the crystalline structure. Tamai et al.[28] showed that the local electronic structure around Sr in $SrFeO_{3-\delta}$ obviously changed with the concentration of oxygen vacancies. These results highlight the essential role of A-site cation in modulating the oxygen activity, yet this long-standing issue have been overlooked to some content, which renders precious control of the oxygen activity remains a grand challenge due to the limited understanding in its activation mechanism.

Herein, we show that the A-site La cation, even if located at the subsurface site beneath the surface Fe-O layer, can exert great influence on the redox performance of $LaFeO_3$. Removing the subsurface La cation ($La_{sub.}$) greatly enhances oxygen mobility and lowers the electron density over O-atom due to the elimination of the $La_{sub.}$-O interaction, which highlights the essential role of redox-inert A-site cation in affecting oxygen activity. To exemplify the practicability of this hypothesis, chemical looping anaerobic oxidation of methane was applied as a probe reaction, wherein the lattice oxygen of metal oxides (known as oxygen carrier) was directly used for C-H bond activation and methane oxidation, substantially giving value-added syngas ($H_2$/CO ratio of 2)[29–31]. It was found that both in-situ redox treatment of stoichiometric $LaFeO_3$ and lowering La/Fe ratio could generate subsurface La vacancies ($La_{vac.}$), enabling pronounced methane

conversion that is 2.9 times higher than stoichiometric $LaFeO_3$ and exhibits excellent syngas selectivity (98%) and cyclic stability.

## Results

### Characterization of fresh redox catalysts

To investigate the role of A-site cation in adjusting the oxygen activity of perovskite oxides, a series of $La_xFeO_3$ (x = 1.03, 1, and 0.97) catalysts with different La content was prepared. XRD results (Fig. S1) show that all reflection peaks of fresh samples are well indexed to the perovskite phase while HRTEM images (Fig. 1b–g) displayed well-resolved lattice fringes with homogeneous mapping of La, Fe, and O elements, which signifies the high purity and fine perovskite-type structure of these oxides. SEM images (Fig. S2) revealed that the particle size notably increased with reducing La content from 50 to 150 nm for $La_{1.03}FeO_3$ to 170–400 nm for $La_{0.97}FeO_3$, corresponding to a decreased specific surface area from 9.4 to 2.2 $m^2$/g.

The oxygen mobility of fresh catalysts was assessed by $O_2$-TPD (Fig. S3) and $H_2$-TPR (Fig. 1h) experiments. During the $O_2$-TPD process, no signals assigned to $O_2$ was detected. This should be induced by the low specific surface areas of these OCs, leading to the trace amount of $O_2$ being adsorbed on surface oxygen vacancies. Besides, the lattice oxygen in $LaFeO_3$ is stable due to the high oxygen vacancy formation energy (>3.62 eV)[21]. Therefore, the desorption of lattice oxygen to generate oxygen vacancy by simple thermal treatment in He is difficult. As for reduction in hydrogen (Fig. 1h), in general, two $H_2$ consumption peaks, located at low-temperature zone from 300 to 500 °C and high temperature above 700 °C, respectively, could be observed over three samples. As for $La_{1.03}FeO_3$ and $LaFeO_3$ samples, the $H_2$ uptakes at low-temperature zone is low, which should be assigned to the reduction of surface-active oxygen species. In contrast, an obviously enhanced reduction peak is detected for the $La_{0.97}FeO_3$ sample, suggesting that partial lattice oxygen is activated upon the introduction of La vacancies into the perovskite matrix. The accumulated $H_2$ consumption for $La_{0.97}FeO_3$ (La-deficiency of 126 μmol/g) at this zone reaches 190 μmol/g, corresponding to $O_{activated}$/$La_{vac.}$ ratio of 1.51, which highlights the important role of La vacancies in activating the lattice oxygen. Compared with the distinct difference at low-temperature

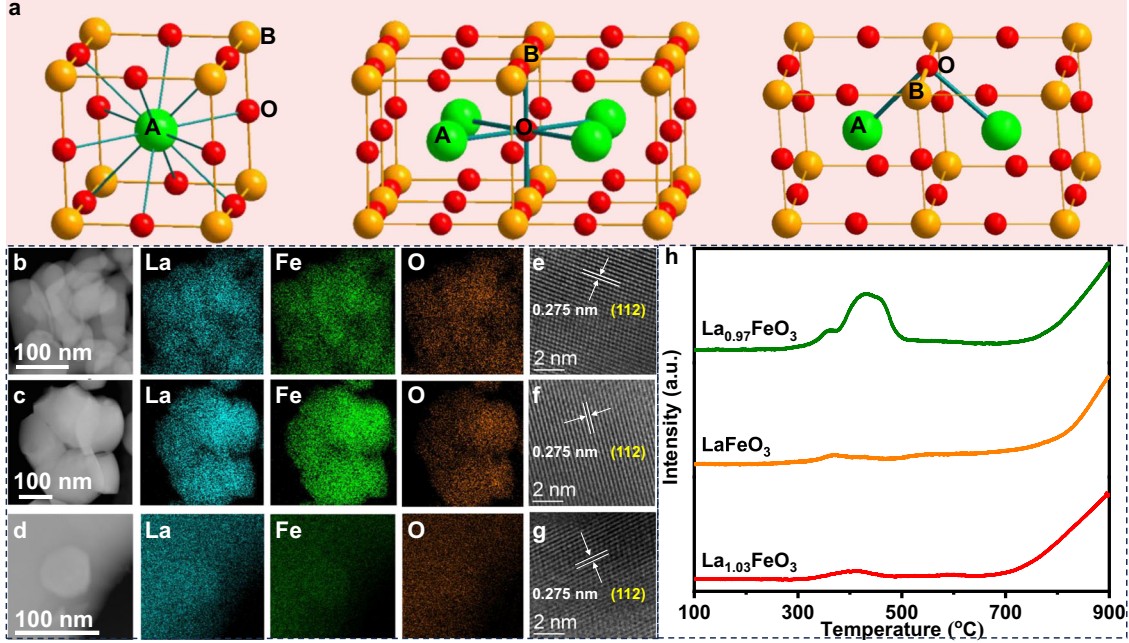

**Fig. 1 | Perovskite structure and characterization of fresh samples. a** Unit cell of perovskite oxide ($ABO_3$) with an ideal cubic structure (left) and corresponding coordination state of bulk (middle) and surface (right) lattice oxygen (perovskite surface terminated with Fe cations). EDS maps of **b** $La_{1.03}FeO_3$, **c** $LaFeO_3$, and **d** $La_{0.97}FeO_3$, and corresponding HRTEM images of **e** $La_{1.03}FeO_3$, **f** $LaFeO_3$, and **g** $La_{0.97}FeO_3$. **h** $H_2$-TPR profiles of $La_xFeO_3$ (x = 1.03, 1, and 0.97) oxides.

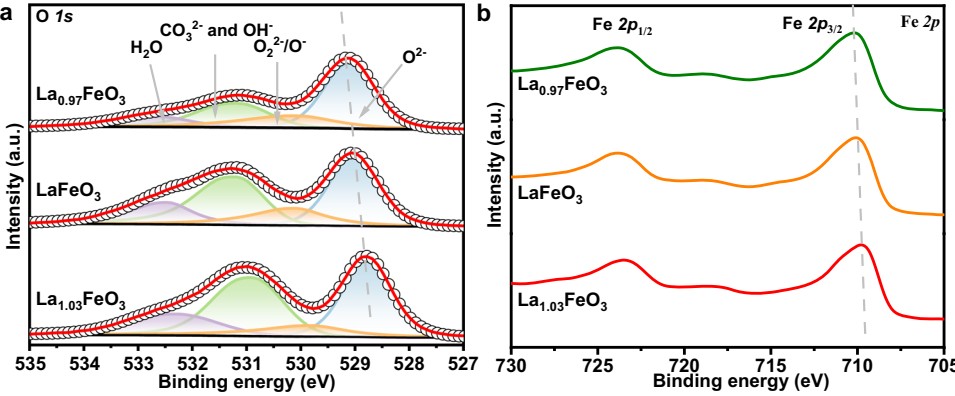

**Fig. 2 | XPS spectra of fresh samples. a** O *1s* and **b** Fe *2p* XPS spectra of La$_x$FeO$_3$ (x = 1.03, 1, and 0.97) oxides.

zone, the reduction behavior of these samples at high temperature is almost similar, which should be induced by the analogous coordination structure of most oxygen atoms in the catalysts.

XPS analysis was conducted to evaluate the chemical state of surface O and Fe. As shown in Fig. 2a, the O *1s* spectra can be deconvoluted into four secondary peaks, which were assigned to lattice oxygen O$^{2-}$ (528.8–529.1 eV), surface adsorbed oxygen O$_2^{2-}$ and O$^-$ (529.8–530.2 eV), carbonate CO$_3^{2-}$ and hydroxyl OH$^-$ (530.9–531.2 eV), and molecular water (532.4–532.7 eV), respectively[32]. It is noted that the proportion of CO$_3^{2-}$ and OH$^-$ notable reduced from 36.8% of La$_{1.03}$FeO$_3$ to 24.6% of La$_{0.97}$FeO$_3$, which coincides with the reduced concentration of alkaline La cations at the surface (Table S1) that leads to degraded adsorption of CO$_3^{2-}$ and OH$^-$. Besides, the peak position of lattice oxygen gradually shifted to higher bonding energy with lowering the concentration of La$^{3+}$ cations, suggesting that the formation of La vacancies renders a declined electron density over the lattice oxygen. Correspondingly, an obvious shift of Fe $2p_{3/2}$ peak to higher binding energy from 709.7 eV of La$_{1.03}$FeO$_3$ to 710.1 eV of La$_{0.97}$FeO$_3$ can be noted (Fig. 2b), suggesting that reduction of La content could induce improved valence state of Fe cations to maintain the charge balance in the oxides[33,34]. Overall, the above results illustrate that engineering the La concentration at A-site of perovskite ferrites is effective to alter chemical state of lattice oxygen, which is expected to exert great influence to the redox activity.

## Effect of La engineering on the oxygen activity

CH$_4$-TPR experiments (Fig. 3a−c) was first conducted to evaluate the oxygen activity of these redox catalysts. As for La$_{1.03}$FeO$_3$, signals assigned to H$_2$ and CO was detected at temperatures of 738 and 746 °C, respectively, which corresponds to the C-H bond cleavage of CH$_4$ (CH$_4$ → *C + 2H$_2$) and selective oxidation of C-atom in CH$_4$ by lattice oxygen (*C + O$_L$ → CO + [O$_{vac.}$]). Although bearing lowered specific surface area, it is noted that the onset temperature of syngas over LaFeO$_3$ is notably reduced to 714 °C for the H$_2$ profile and 718 °C for the CO profile, respectively, and further slightly declined to 712 °C for the H$_2$ profile and 717 °C for CO profile over La$_{0.97}$FeO$_3$ sample. Besides, the intensity of syngas signals was also gradually enhanced. These results suggest that lowering La concentration at the A-site could greatly promote the intrinsic reactivity for methane anaerobic oxidation.

To assess the redox performance, these catalysts were subjected to isothermal redox reaction, wherein the catalysts was reduced by CH$_4$ and regenerated by CO$_2$ oxidation. The spent catalyst was marked as La$_x$FeO$_3$-Y (Y represents the number of redox cycles). As deposited in Fig. 3d−f, upon feeding CH$_4$ to these catalysts, obvious signals assigned to syngas with an H$_2$/CO ratio of ca. 2.0 (Fig. S4) were observed immediately over all samples while signals of CO$_2$ are negligible, which highlights the outstanding selectivity of perovskite ferrites

towards syngas production. The syngas productivity (Fig. 3g) in the first redox cycle was gradually improved with the order of La$_{1.03}$FeO$_3$ (2.7 mmol/g) <LaFeO$_3$ (4.7 mmol/g) <La$_{0.97}$FeO$_3$ (5.2 mmol/g), which is in line with CH$_4$-TPR results. Although these catalysts remain a high syngas selectivity above 95% in subsequent cycling tests (Fig. S4), the reactivity differs greatly. As for La$_{1.03}$FeO$_3$-3 and LaFeO$_3$-3, the activity notably reduced by 70% (0.8 mmol/g) and 63% (1.7 mmol/g), respectively, in the first three cycles. In contrast, the syngas productivity of La$_{0.97}$FeO$_3$-3 sample only slightly reduced by 9%, highlighting the promoting effect of La-deficiency in improving the redox performance (Fig. 3h). Besides, it is surprising to note that the reactivity of LaFeO$_3$ is gradually recovered from 4th cycle to 20th cycle, substantially giving a steady syngas yield (4.6 mmol/g) in the following reaction until 250th cycle (Fig. 3h, i). A similar variation in CO$_2$ conversion was also observed as shown in Fig. S5. To double-check this phenomenon, CH$_4$-TPR experiments were conducted for fresh LaFeO$_3$ and samples after three cycles and 20 cycles (Fig. S6). It was found that methane conversion was greatly suppressed over LaFeO$_3$-3, as suggested by the reduced intensity for syngas production, while the reactivity of LaFeO$_3$-20 was almost identical to that of fresh LaFeO$_3$. These results suggest that surface reconstruction may occur for LaFeO$_3$ samples during redox cycles, which induces the high reactivity matching that of La$_{0.97}$FeO$_3$.

## Structure evolution of catalysts during isothermal redox reactions

To better understand the parameters that determine the oxygen activity, the catalysts after isothermal redox reactions are subjected to detailed characterizations. The XRD results (Fig. S7) show that the perovskite structure is maintained for catalysts after 20 redox cycles, which suggests that no obvious phase separation that gives nano-sized La$_2$O$_3$ or FeO$_x$ as impurities occurs within this reaction period. The SEM results (Fig. S8) demonstrate that the particle size of La$_{1.03}$FeO$_3$-20, LaFeO$_3$-3, and La$_{1.03}$FeO$_3$-20 in these samples all notably increased after undergoing oxidation-reduction treatment, which should account for the seriously degraded reactivity of La$_{1.03}$FeO$_3$ and LaFeO$_3$-3 samples. However, for LaFeO$_3$, as the number of cycles was extended to 20 or even 250 cycles, the growth of particle size slowed down (Fig. S8). Besides, it is noted that samples with higher La content maintain a smaller particle size after 20 redox cycles, indicating that surface La enrichment is beneficial for inhibiting the sintering of oxygen carrier particles. To further disclose the distinct redox performance of these samples, XPS analysis was conducted to monitor the surface structure evolution. As deposited in Fig. 4, no obvious changes were observed for O *1s* and Fe *2p* profiles of fresh La$_{1.03}$FeO$_3$ and La$_{1.03}$FeO$_3$-20, which indicates that surface reconstruction during redox cycles was negligible over the La-excess sample. In contrast, compared with the fresh samples, the ratio of

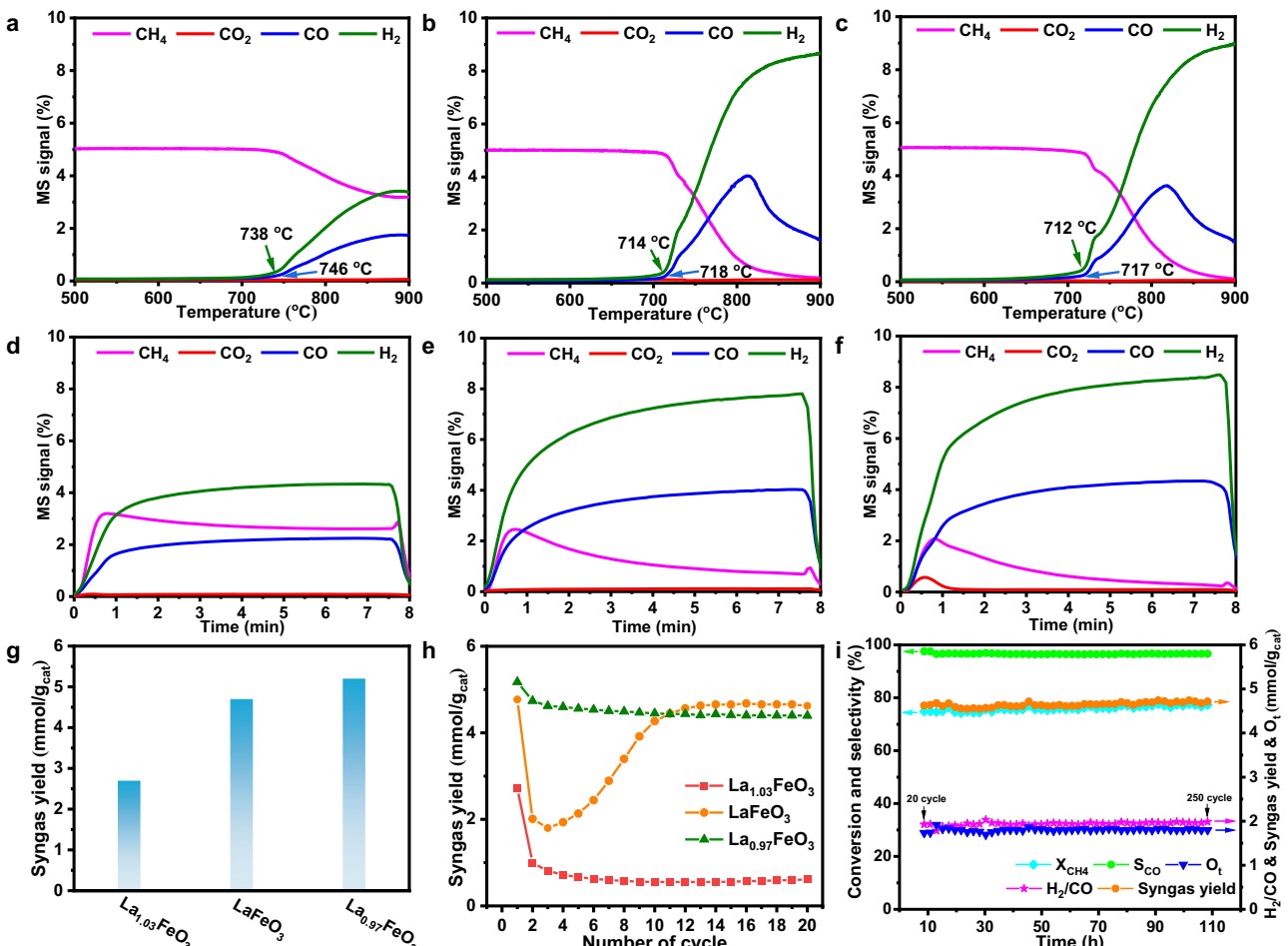

**Fig. 3 | Catalytic performance and stability test.** CH₄-TPR profiles for **a** La₁.₀₃FeO₃, **b** LaFeO₃, and **c** La₀.₉₇FeO₃. Reaction conditions: 100 mg catalyst treated with 5% CH₄/He (30 mL/min) from 20 to 900 °C with a rate ramp of 10 °C/min. The kinetic curves for the methane partial oxidation step in the first cycle of **d** La₁.₀₃FeO₃, **e** LaFeO₃, and **f** La₀.₉₇FeO₃. **g** The syngas yields of fresh LaₓFeO₃ (x = 1.03, 1 and 0.97) samples. **h** The syngas productivity during 20 redox cycles for LaₓFeO₃ samples (x = 1.03, 1, and 0.97). **i** The performance of CH₄ partial oxidation step for LaFeO₃ from 20th cycle to 250th cycle (8–108 h). Reaction conditions: 100 mg catalyst treated with 5% CH₄/He (15 mL/min) for 8 min during CH₄ partial oxidation step, 5% CO₂/He (15 mL/min) for 10 min during CO₂ regeneration step at 900 °C, and the reactor was purged with He for 4 min (20 ml/min) between partial oxidation and reoxidation step.

surface carbonates notably improved by 15.8 and 59.6% for LaFeO₃-20 and La₀.₉₇FeO₃-20 samples (Table S1). Consequently, a slight shift of Fe 2*p* spectra to higher binding energy is observed. These results give a clue that partial La cations should be removed from the perovskite matrix and aggregates into oxide clusters during the redox reaction.

To double-check the above conclusion, the surface structure evolution of LaFeO₃ during redox reaction was analyzed by LEIS, which is sensitive to the changes of the first layer atoms at the catalyst surface[35,36]. As shown in Fig. 5a and Fig. S9, the surface of fresh LaFeO₃ is composed of 80% La cations and 20% Fe cations. This suggests that LaFeO₃ is mainly terminated by La cations at A-site, which is in line with previous reports[37]. After three redox cycles, it is noted that the ratio of surface La notably reduced to 42%, rendering the first layer atoms mainly terminated by Fe cations (58%). This indicates that the redox reaction can result in aggregation of the outmost La cations, as elucidated in Fig. 5a, leading to Fe as the overwhelming cation at the surface. It is interesting to note that the ratio of surface La further increased to 50% after 20 cycles, suggesting that the continuous redox treatment can induce exsolution of more La cations (La_sub.) beneath surface Fe out of perovskite matrix. This conclusion can be also supported by the AC-HAADF-STEM result (Fig. 5b-left) of LaFeO₃-20. The site with weakest La-atom intensity (marked by black arrow), where accommodating the fewest number of La-atoms, from first layer to

fourth layer (Fig. 5b-right) differs greatly with each other. This result suggests that La cation vacancies were generated and randomly distributed at these layers. Besides, XRD results show that obvious signals assigned to La₂O₃ can be detected after 50 cycles, which should be induced by the sustained out-diffusion of La cations from the subsurface sites. Corresponding XRD and Raman analysis for LaFeO₃-20 and LaFeO₃-50 (Fig. S10) shows that phases of FeOₓ is not observed in these spent catalysts, indicating that the Fe cations are still stabilized in perovskite matrix rather than aggregates into FeOₓ nanoparticles after removal of subsurface La cations[38–40]. Overall, these results demonstrate that subsurface La ions in stoichiometric LaFeO₃ can migrate out of the perovskite matrix during the redox reaction, which results in surface enrichment of La₂O₃ oxide and formation of subsurface La vacancies, substantially contributing to the sustained redox performance despite of gradually increased particle size of these catalysts.

The inherent reason for generating subsurface La vacancies was studied by analyzing the structural evolution of LaFeO₃ during CH₄-CO₂ redox treatment. It was found that, as shown in Fig. S11, when LaFeO₃ was reduced by CH₄ (marked as CH₄-Re), some Fe cations were reduced to metallic Fe⁰ with a production of La₂O₃ oxides. Subsequent oxidation by CO₂ could recover the lattice oxygen. After 4 min of oxidation (marked as CO₂-Ro-4min), it was found that no signals for

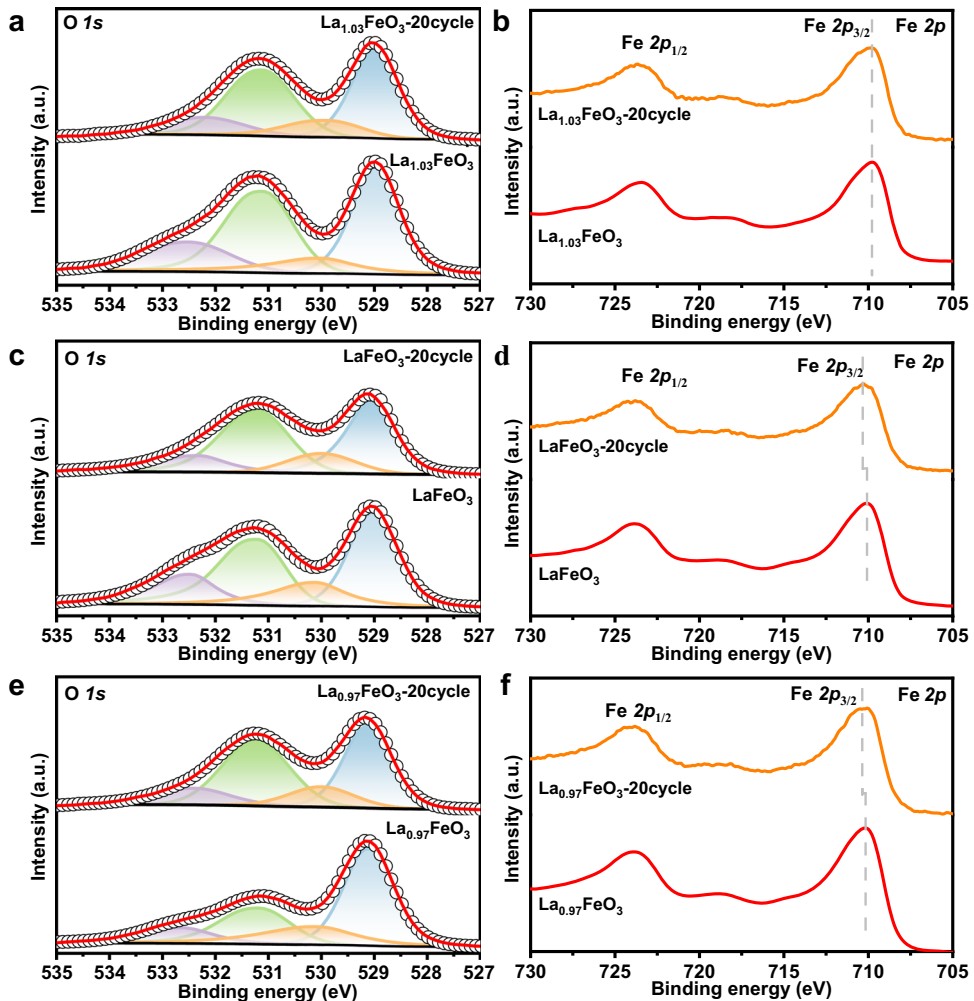

**Fig. 4 | XPS spectra of fresh and cycled samples. a, b** O *1s* and Fe *2p* for La$_{1.03}$FeO$_3$ and La$_{1.03}$FeO$_3$-20, **c, d** O *1s* and Fe *2p* for LaFeO$_3$ and LaFeO$_3$-20, **e, f** O *1s* and Fe *2p* of La$_{0.97}$FeO$_3$ and La$_{0.97}$FeO$_3$-20.

Fe$^0$ (or FeO$_x$) could be observed while that of La$_2$O$_3$ can be clearly detected, suggesting that the access of Fe cations back to LaFeO$_3$ perovskite matrix proceeds much faster than that of La cations. This is mainly due to the slow rate of A-site cation migration (cation size is much larger than that of B-site cation), a rate-controlling step for the formation of an ideal perovskite crystal structure with an A/B ratio of 1[41,42], which induced the formation of subsurface La vacancies in LaFeO$_3$ during chemical looping processes.

**Theoretical insights into the effect of subsurface La on oxygen activity**

To gain insight into the effect of La cation engineering on oxygen activity, density functional calculations (DFT) were further conducted. There are two different possible surface structures for LaFeO$_3$: the Fe-O terminated surface and the La-O terminated surface (Fig. S12). As shown in Table S2, although the La-O terminated surface is relatively more stable, it is almost inert for the methane partial oxidation reaction[43]. Considering the high performance of lanthanum ferrite, the Fe-O terminated surface was used as the model structure due to its much better activity than the La-O terminated surface for methane conversion. As for the location of La vacancies, there are four different sites could be identified (Fig. S13), i.e., La1$_{vac}$, La2$_{vac}$, La3$_{vac}$, and La4$_{vac}$, because of the two different O sites in the LaFeO$_3$ structure, i.e., top O (coordinated with La1 and La2) and bottom O (coordinated with La3 and La4). According to the structural symmetry, La1$_{vac}$ and La3$_{vac}$ are

equivalent to La2$_{vac}$ and La4$_{vac}$, respectively. As tabulated in Table S3, the total energy of LaFeO$_3$ with La1$_{vac}$ and total energy with La3$_{vac}$ are almost the same, indicating single La vacancy is possibly generated in both La1$_{vac}$ and La3$_{vac}$ sites. However, the top O shows lower O vacancy formation energy than the bottom O in a single La vacancy structure for LaFeO$_3$, suggesting that the top O is conducive to activate methane (Table S4). Consequently, the LaFeO$_3$ with La1$_{vac}$ and La2$_{vac}$ was used as the models for the following calculations. Additionally, we present the results calculated based on the bottom O (Figs. S14, 15 and Table S5) for better reference.

Deformation of surface structure with a concentration of La$_{sub.}$ cations were first studied and displayed in Fig. 6a−c. It is found that the Fe$_{surface}$-O-Fe$_{surface}$ motif (coordinated with the top O) bears a bond angle of 171.6° with an average Fe-O bond length of 1.930 Å for stoichiometric LaFeO$_3$ with two La$_{sub.}$ cations located beneath the surface oxygen. Once one La$_{sub.}$ was removed, out-of-plane movement occurs to the oxygen due to degraded La$_{sub.}$-O interaction, leading to a decreased Fe$_{surface}$-O-Fe$_{surface}$ bond angle to 153.9° with reduced average Fe-O bond length of 1.880 Å. As for oxygen without adjacent La$_{sub.}$ (two La$_{vac.}$), the bond angle further reduced to 126.1° while the average Fe-O bond length reduced to 1.806 Å. These results suggest that the La$_{sub.}$ can exert great influence on the chemical state of surface Fe and O.

The evolution of La coordination on Fe$_{surface}$-O interaction was verified by the analysis of the total density of state (TDOS), charge

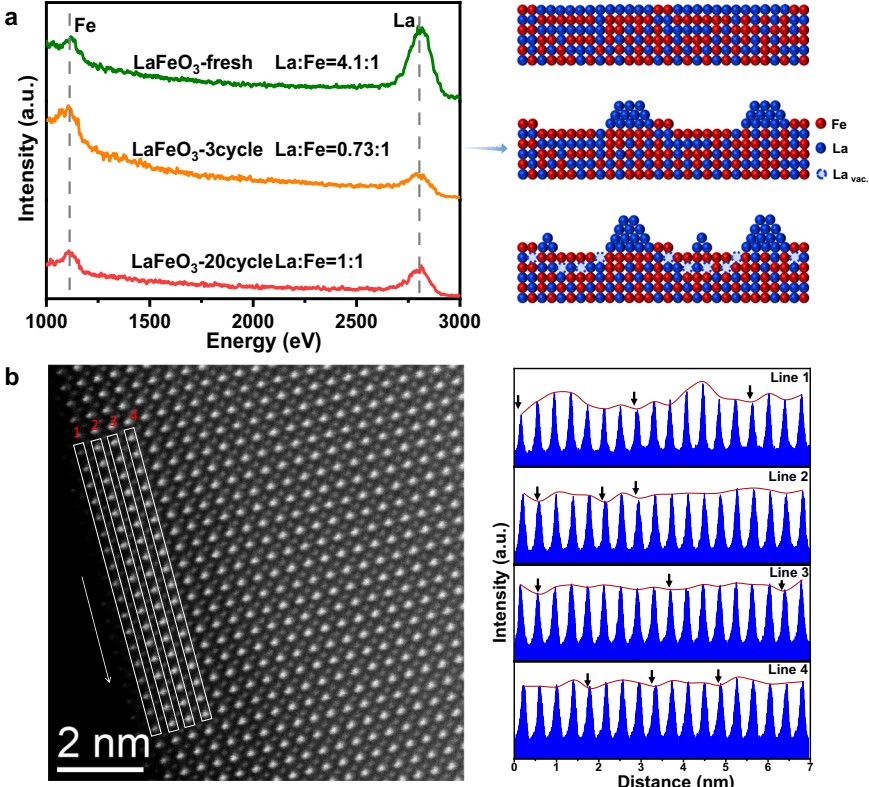

**Fig. 5 | Characterization of subsurface La vacancies. a** The LEIS results for La and Fe atoms in the first layer of La$_x$FeO$_3$-Y oxides and corresponding schematic representation of the surface structural evolution. **b** Aberration-corrected high-angle annular dark-field scanning transmission electron microscopy image of LaFeO$_3$-20 along with corresponding showcasing diagram of La content in first four layers. (La$_{vac.}$ stands for subsurface La vacancy).

transfer energy from O $2p$ to Fe $3d$ band, and the Bader charge. As shown in Fig. 6d, the overlap of Fe $3d$ and O $2p$ orbitals in energy interval from −8 to 0 is detected, which is characteristic of the covalent feature of Fe-O bond. Corresponding charge-transfer energy was calculated based on the center of Fe and O bands[44,45]. It was found that the charge-transfer energy (Fig. 6e) was notably reduced from 4.87 eV (no La$_{vac.}$) to 4.31 eV (two La$_{vac.}$ sites) with increasing the La vacancies, highlighting the enhanced Fe-O covalency due to weakened La$_{sub.}$-O interaction. Consequently, the Bader charge (Table S6) of Fe was gradually increased from +1.64 (no La$_{vac.}$) to +1.67 (two La$_{vac.}$ sites), while that of O slightly decreased from −1.11 (no La$_{vac.}$) to −1.02 (two La$_{vac.}$ sites), which is in line with the XPS analysis (Fig. 3). Oxygen mobility is one key parameter that determines the redox properties of metal oxides. To quantitatively evaluate the influence of subsurface La cations on oxygen mobility, the variation of oxygen vacancy formation energy ($E_v$) with the number of La$_{sub.}$ was calculated. As for stoichiometric LaFeO$_3$, $E_v$ of 3.25 eV needs to be conquered to remove the surface lattice oxygen coordinated with two Fe$_{surface}$ and two La$_{sub.}$ cations. When one La$_{vac.}$ exists, the value of $E_v$ is notably reduced to 1.37 eV, which is further lowered to 0.79 eV for lattice oxygen with two adjacent La$_{vac.}$ sites (Fig. 6f–h). These results demonstrate that it is indeed the intimate La$_{sub.}$-O interaction, instead of Fe$_{surface}$-O, plays a dominant role in determining the $E_v$ value and renders a high stability of surface oxygen. Therefore, generating subsurface La vacancies would greatly improve the surface oxygen mobility.

The above results demonstrate that engineering the concentration of La cations at the subsurface is efficient in modulating the oxygen mobility and the Fe$_{surface}$-O interaction, which would substantially influence the catalytic performance. Such evolution of the oxygen state on the methane activation is then evaluated. It was found that the H-atom of CH$_4$ prefers to adsorb on the lattice oxygen, while

the generated *CH$_3$ is stabilized by the adjacent Fe cation. The adsorption energy H-atom, one main driving force of methane activation, notably increased with removing the La$_{sub.}$ cations from −1.43 eV (no La$_{vac.}$) to −2.04 eV (two La$_{vac.}$ sites), as seen in Table S6, suggesting that the reduction of electron density over O-atom can intensify the hydrogen adsorption process. Consequently, homolytic cleavage of the C-H bond, generally accepted as the rate-determining step for CH$_4$ conversion, is preferred over La-deficient catalysts and the activation barrier notably reduced from 1.88 eV (no La$_{vac.}$) to 1.03 eV (two La$_{vac.}$ sites), as deposited in Fig. 6i and Table S7[46,47]. Besides, it was found that the energy barriers for cleavage of other C-H bonds in *CH$_3$ or generation of syngas (H$_2$ and CO) are obviously reduced after generating subsurface La vacancies, which should be ascribed to the reduced electron intensity over lattice oxygen and the declined oxygen vacancy formation energy (Fig. 6i and Table S7). Overall, the calculation results signify that the generation of subsurface La vacancies represents an effective method for altering the redox performance of perovskite catalysts and can potentially promote oxygen reactivity for methane conversion.

## Discussion

According to previous studies, selective anaerobic oxidation of CH$_4$ by redox catalysts proceeds through C-H bond cleavage (CH$_4$ → *C + 2H$_2$) and oxidization of C-atom by lattice oxygen (*C + O$_L$ → CO + [O$_{vac.}$]), substantially giving H$_2$ and CO as the products[21]. Selection of suitable redox catalysts with balanced activity for breaking the C-H bond and oxidizing the deposited C-atom is the key to an efficient methane-to-syngas process[5,48,49]. The B-O redox pair in perovskite catalysts was generally regarded as the active centers for methane conversion, triggering a strong passion to promote the redox performance by terminating catalyst surface by more B-O pair

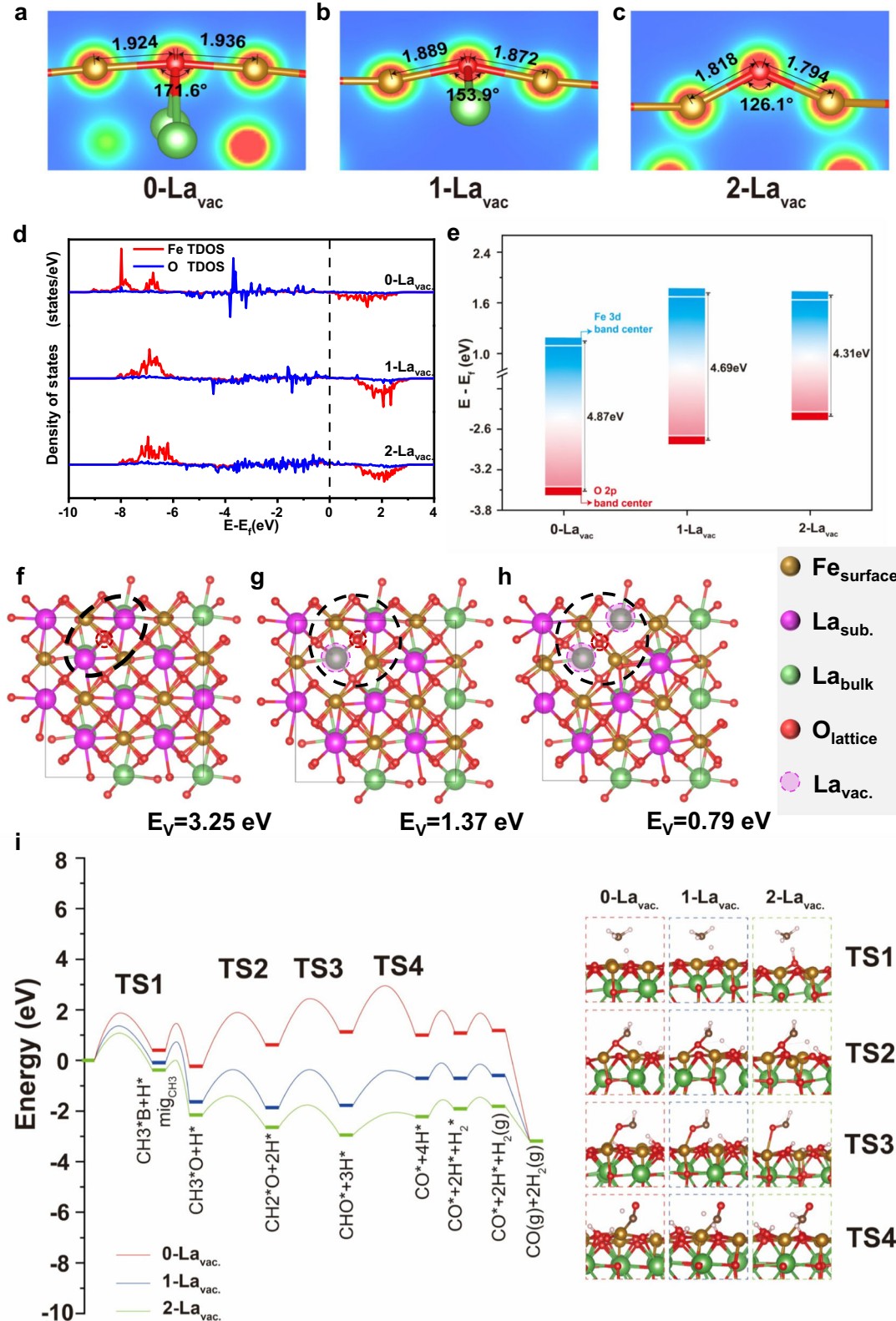

**Fig. 6 | Theoretical investigations on the effect of subsurface La vacancies.**
**a–c** Surface structure changes with La$_{sub.}$ vacancy concentration, **d** TDOS of Fe and O bands for different La$_{sub.}$ vacancy concentration, **e** charge-transfer energy with corresponding band centers of unoccupied Fe $3d$ and occupied O $2p$ states for different La$_{sub.}$ vacancy concentration, **f–h** computational model of oxygen vacancy formation in the bulk of different La$_{sub.}$ vacancy concentration, and **i** comparison of energy profile for CH$_4$ activation over oxygen coordinated with different number of La$_{sub.}$ vacancies. La$_{vac.}$ stands for subsurface La vacancy; La$_{sub.}$ stands for subsurface La.

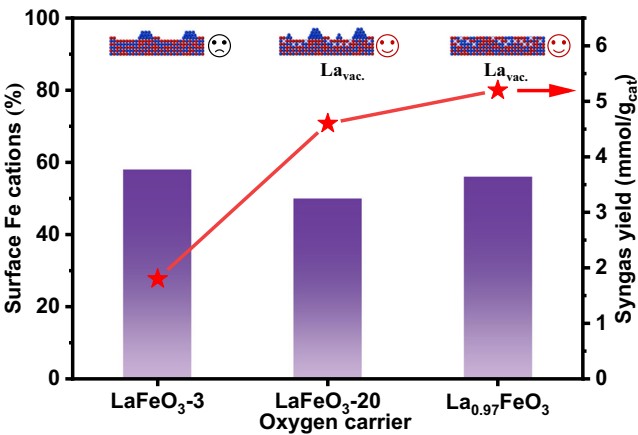

**Fig. 7** | Correlation between surface Fe percentage and corresponding methane anaerobic oxidation performance for LaFeO$_3$-3, LaFeO$_3$–20, and La$_{0.97}$FeO$_3$. La$_{vac.}$ stands for subsurface La vacancy.

(instead of A-O) or regulating the B-O interaction via asymmetric B$_1$-O-B$_2$ interaction.

As for lanthanum ferrite, the redox capability of the Fe-O pair is superior to the La-O pair, rendering the catalysts terminated mainly by Fe cations more active for methane conversion. This also explains the low reactivity of the La-excess La$_{1.03}$FeO$_3$ sample for methane anaerobic oxidation. However, it is noticed that the samples of LaFeO$_3$–3, LaFeO$_3$-20, and La$_{0.97}$FeO$_3$, although bearing similar percentages of Fe cations at the surface and specific surface area, display distinct performance. After three redox cycles, the percentage of surface Fe cations increased from 20% (fresh LaFeO$_3$) to 58% (LaFeO$_3$-3), while the syngas yield notably declined from 4.7 mmol/g (LaFeO$_3$) to 1.8 mmol/g (LaFeO$_3$-3), which was mainly induced by the notable decline of BET surface area (from 5.8 of LaFeO$_3$ to 1.3 m$^2$/g of LaFeO$_3$-3) (Fig. S16 and Table S8). When prolonging the tests to 20 cycles, substantial change in specific surface area was insignificant (from 1.3 m$^2$/g of LaFeO$_3$-3 to 1.8 m$^2$/g of LaFeO$_3$–20), while a slight decline of surface Fe percentage from 58 to 50% occurs due to continuous removal of La cations from the perovskite matrix (Fig. 7). However, the syngas yield was notably improved by 2.5 times (from 1.8 mmol/g of LaFeO$_3$-3 to 4.6 mmol/g of LaFeO$_3$-3). Meanwhile, the La$_{0.97}$FeO$_3$ sample with 3% La-deficiency that bears 56% surface Fe cations (Fig. S17) and a similar surface area of 2.2 m$^2$/g gives a syngas yield of 5.2 mmol/g, which is 2.9 times higher than the LaFeO$_3$-3 sample. Combined with the experimental studies and DFT calculations, it is found that subsurface La vacancies can be generated by conducting redox treatment, such as the chemical looping process, or by lowering the La/Fe ratio. The absence of subsurface La (electron donor), on the one hand, reduces the electron density over surface oxygen (Table S6), which is favorable for hydrogen (from CH$_4$) adsorption, lowering the energy barrier for homolytic cleavage of the C-H bond. On the other hand, the absence of strong La$_{sub.}$-O interaction greatly promotes oxygen mobility, which contributes to oxidizing the C-atom of CH$_4$ to CO, substantially leading to significantly enhanced performance for methane anaerobic oxidation to syngas.

In summary, we propose the finding that generating subsurface La vacancy for lanthanum ferrite can directly tune the electronic state of lattice oxygen, which ultimately activates the lattice oxygen to enhance the redox performance in methane anaerobic oxidation reaction. The syngas yield over the sample with subsurface La vacancy is 2.9 times larger than that of stoichiometric LaFeO$_3$–3. Combined characterizations and DFT calculations show the formation of subsurface La vacancy can lower the electron density over surface lattice oxygen, which induces a facile homolytic cleavage of the C-H bond.

Besides, the absence of La$_{sub.}$-O interaction could reduce the oxygen vacancy formation energy, contributing to improve the oxygen mobility and oxidation of deposited C-atom. Consequently, the methane-to-syngas conversion is notably promoted over the La-deficient lanthanum ferrites. This strategy of generating subsurface A-site vacancies would be useful for designing advanced redox catalysts towards reactions proceeded via the Mars–van Krevelen mechanism and chemical looping processes.

## Methods

### Catalyst preparation
The La$_x$FeO$_3$ (x = 0.97, 1, 1.03) samples were prepared via the combustion method. Taking LaFeO$_3$ as an example, firstly, 0.87 g of La(NO$_3$)$_3$·6H$_2$O, 0.81 g of Fe(NO$_3$)$_3$·9H$_2$O, and 0.60 g of glycine (the molar ratio of citric acid/total metal ions was 2) were dissolved in 1.2 g of water. The solution is then dried in an oven at a constant temperature of 60 °C for 1 h, which were then burned in a muffle furnace for 10 min at 500 °C. Finally, the obtained powders were crushed into powder and calcined at 850 °C for 4 h.

### Characterization
The X-ray diffraction (XRD) patterns were acquired using a PANalytical X'Pert-Pro powder X-ray diffractometer (U = 40 kV, I = 40 mA) with Cu Kα radiation of λ = 0.15418 nm to analyze the crystal structure and phase composition of fresh samples. The Raman spectrum was collected by a Raman-atomic force microscope imaging analysis system with an incident wavelength of 532 nm and a scan range of 100–500 cm$^{-1}$. The morphology of these catalysts was observed by field emission scanning electron microscopy (JEOL JSM-7800F SEM). The high-resolution transmission electron microscopy (HRTEM) images and elemental distribution of the samples were obtained by transmission electron microscopy (JEOL JEM-2100F TEM). Atomic column high-angle annular dark-field scanning transmission electron microscopy (AC-HAADF-STEM) analysis was conducted using a JEOL JEM-ARM200F microscope fitted with a CEOS probe corrector to achieve a resolution of 0.08 nm. The specific surface area was obtained by calculating the results of the QUADRSORB SI analyzer using the Brunauer–Emmett–Teller (BET) method. Low energy ion scattering (LEIS) was conducted on an ION-TOF Qtac$^{100}$ for surface compositional analysis and depth profiling. Before the analysis, the sample was treated at a temperature of 300 °C for 1 h to remove the surface contaminant, and the charge neutralization technique was used for spectral acquisition and sputtering. With Ne$^+$ as the detection source, different sputtering depths were calculated by adjusting the sputtering time. X-ray photoelectron spectroscopy (XPS) was tested on a Thermo Fisher Nexsa-type photoelectron spectrometer. To avoid the influence of air exposure on the analysis results, the samples after the chemical looping reaction (oxidized by CO$_2$) were cooled down to room temperature in He atmosphere, which were then moved into a well-closed XPS transfer bin in the glove box (Ar protected). The elemental composition and ionic state of the surface samples were analyzed using a monochromatic Al target Kα, and the binding energy was calibrated according to the contaminated carbon-C 1s (284.8 eV). The oxygen activity of fresh samples was studied by H$_2$ temperature-programed reduction (H$_2$-TPR) over Micromeritics Autochem II 2920 apparatus. Specifically, 200 mg of sample was placed in a quartz tube and the temperature was gradually increased to 900 °C at a rising rate of 10 °C /min in a 10% H$_2$/Ar mixture (50 mL/min). The CH$_4$-TPR experiments were carried out in a fixed bed reactor with 100 mg of 60–80 mesh catalyst in a 6 mm quartz tube, ramped at 10 °C/min until 900 °C in 5% CH$_4$/He (30 mL/min).

### Isothermal redox reactions
Experiments to evaluate the redox activity and stability of these catalysts were carried out on an atmospheric pressure fixed bed reactor.

Typically, the fresh sample of 100 mg (60–80 mesh) was placed in a quartz tube with an inner diameter of 6 mm. A successive $CH_4$ reduction-$CO_2$ regeneration redox cycles were performed at 900 °C. The catalyst were reduced in a 5% $CH_4$/He atmosphere for 8 min (15 mL/min) before being regenerated in a 5% $CO_2$/He atmosphere for 10 min (15 mL/min). Between reduction and reoxidation reaction, the reactor was purged with He for 4 min (20 ml/min). In addition, the products were analyzed using an online quadrupole mass spectrometer (IPI GAM 200). Before the isothermal redox reaction, a standard gas with known composition is used to calibrate the mass spectrometer. The average $CH_4$ conversion ($X_{CH4}$), CO selectivity ($S_{CO}$), syngas yield ($Y_{syngas}$), oxygen output ($O_t$), $H_2$/CO ratio ($R$) for methane anaerobic oxidation step, and $CO_2$ conversion ($X_{CO2}$) for $CO_2$ regeneration step was calculated by following equations:

$$X_{CH_4} = \frac{n_{CH_{4-in}} - n_{CH_{4-out}}}{n_{CH_{4-in}}} \tag{1}$$

$$S_{CO} = \frac{n_{CO-out}}{n_{CH_{4-in}} - n_{CH_{4-out}}} \tag{2}$$

$$Y_{syngas} = \frac{n_{CO} + n_{H_2}}{m} \tag{3}$$

$$O_t = \frac{4n_{CO_{2-out}} + n_{CO_{out}}}{m} \tag{4}$$

$$R = \frac{n_{H_2}}{n_{CO}} \tag{5}$$

$$X_{CO_2} = \frac{n_{CO_{2-in}} - n_{CO_{2-out}}}{n_{CO_{2-in}}} \tag{6}$$

The term $n_x$ represents the amount of the corresponding gas x (mol) flowing into or out of the reactor, and $m$ is the mass of the catalyst used.

## Density functional theory calculations

We used the density functional theory (DFT) calculations with the Vienna ab initio simulation package (VASP) software. For the core electrons, we utilized the projector augmented wave (PAW) method and the Perdew–Burke–Ernzerhof (PBE) exchange-correlation function to describe them[50]. The wavefunctions were expanded on a plane-wave basis, corresponding kinetic energy cutoff was set to 400 eV. We used a $3 \times 3 \times 1$ gird to employ the gamma-centered k-point mesh. To deal with the strong Coulomb interactions of Fe $3d$ electrons, we carried out the DFT + U method to revise it[51]. The geometric optimization was known as convergent only when the energy change was smaller than 0.02 eV Å$^{-1}$. Also, only the energy change was less than $10^{-5}$ eV, and the electron energy was considered self-consistent. When the energy and force were reached to the expected outcome simultaneously, the energy of optimized structures were considered obtained. To avoid the interaction between the periodic images in the z-direction, a vacuum gap of 15 Å was set. Apart from the top two layers, the others were fixed to ensure the geometry to be constant during the optimization process. The adsorption energy ($E_{ads}$) for adsorbate can be calculated in the form below equation:

$$E_{ads} = E_{ad/sub} - E_{ad} - E_{sub} \tag{7}$$

$E_{ad/sub}$ is the electronic energy of the optimized adsorbate/substrate system. The $E_{ad}$ and $E_{sub}$ denote the energies of the sole adsorbate and the clean substrate, respectively. Moreover, vacancy formation energy ($E_{vac}$) was determined by equation:

$$E_{vac} = E_{defect-surface} - E_{O2} - E_{surface} \tag{8}$$

where $E_{defect-surface}$, $E_{O2}$, and $E_{surface}$ represent the total energies of the defective surface system, the $O_2$, and the complete surface system, respectively. The total orbital band center $\varepsilon_{tot}$ was obtained through equation:

$$\varepsilon_{tot} = \frac{\int_{-\infty}^{+\infty} n_{tot}(\varepsilon)\varepsilon d\varepsilon}{\int_{-\infty}^{+\infty} n_{tot}(\varepsilon) d\varepsilon} \tag{9}$$

where $n_{tot}(\varepsilon)$ and $\varepsilon$ indicate the value of the total orbital density of states (TDOS) and the energy of the total orbital. To find the ion's relative valence, we calculated the Bader charge, and then through normalizing the number of valence electrons, we can obtain the relevant oxidation states. The relative distance between the occupied O $2p$ center and the unoccupied Fe $3d$ center were calculated to represent the dimension of Fe-O covalence, which is also can be proved by the relative size of the Bader charge. The climbing-image nudged elastic band (CI-NEB) method was employed to find the transition states (TS) roughly with a force convergence of 0.5 eV Å$^{-1}$ and an energy convergence of $10^{-5}$ eV. Then, the precise outcomes were searched by the dimer method with a force convergence of 0.05 eV Å$^{-1}$ and an energy convergence of $10^{-7}$ eV.

## Data availability
The data that support the findings of this study are all available within the paper and its Supplementary Information.

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

## Acknowledgements

Financial support from the National Natural Science Foundation of China (grant No. 21706254 (C.H.), 22378331 (Y.Z.), and 22178337 (X.W.)), NSFC Center for Single-Atom Catalysis (grant No. 22388102 (X.W.)), and Youth Innovation Promotion Association, CAS (grant No. 2023189 (C.H.)) are gratefully acknowledged. Furthermore, the authors would like to extend sincere gratitude to Engineer Xiaoli Pan for their invaluable assistance with the electron microscopy work.

## Author contributions

J.H.: Investigation, methodology, data curation, validation, and writing—original draft. T.W.: DFT calculation, formal analysis, and software. X.B.: Data curation and formal analysis. Y.T.: Data curation and formal analysis. C.H.: Conceptualization, formal analysis, supervision, writing—review and editing, funding acquisition, and project administration. W.X.: Methodology and validation. Y.H.: Investigation and formal analysis. Z.W.: Software and investigation. B.J.: Methodology and formal analysis. Y.G.: Formal analysis. Y.Z.: Funding acquisition, methodology, supervision, review, and editing. X.W.: Funding acquisition, project administration, supervision, review, and editing.

## Competing interests

The authors declare no competing interests.
