## [Peer Review File · Nature Communications]

Subsurface A-site vacancy activates lattice oxygen in perovskite ferrites for methane anaerobic oxidation to syngasREVIEWER COMMENTS

Reviewer #1 (Remarks to the Author):

The authors present a comprehensive study on the enhancement of oxygen activity in LaFeO₃ perovskite oxides by creating subsurface La vacancies. Through a combination of experimental methods and theoretical calculations, it demonstrates that removing subsurface La cations significantly improves oxygen mobility and lowers the electron density of oxygen atoms. This leads to a 2.9-fold increase in methane conversion efficiency and high syngas selectivity (98%) in anaerobic oxidation, compared to stoichiometric LaFeO₃. The study challenges conventional understanding and opens new avenues for designing redox catalysts, especially for reactions via the Mars–van Krevelen mechanism and chemical looping processes. I recommend it be published after the following comments are addressed:

1. The authors should provide a more detailed explanation of how the subsurface La vacancies are formed, and how to control the vacancy concentration. This could include the specific conditions under which these vacancies are generated.
2. They found an increase in particle size observed via SEM analysis post-redox cycling. How does this increase in particle size correlate with the catalytic activity, especially considering that larger particles often have a reduced surface area?
3. The manuscript describes the migration of La cations out of the perovskite matrix during redox cycles, leading to surface reconstruction. It is unclear how this process specifically affects the stability.
4. The DFT calculations could be better correlated with the experimental results. The surface free energy of different possible surface structures (different terminated surfaces) should be calculated to confirm the model used in this work is reasonable and stable. At the same time, the effect of La vacancy at different locations, or different combinations of LaFeO₃, needs to be discussed.
5. In addition, to compare the activity of 0-La(vac), 1-La(vac), and 2-La(vac), the energy profile for the whole reaction on these surfaces should be provided.

Reviewer #2 (Remarks to the Author):

Comments:

Whether XPS was carried out in-situ or ex-situ? If it is ex-situ then there is possibility of oxidizing the oxides during exposure to air and it will mislead the real analysis. Temperature programmed surface reaction (TPSR) and O₂-TPD is necessary to find out the oxygen vacancy generation and their correlation. Authors can also check whether generating subsurface La vacancy the methane activation temperature is reduced or not? In my opinion this catalyst can activate methane at higher temperature. What about the time-on-stream effect? The authors should check the activity for at least 100 h time-on-stream.

Response to reviewers

GENERAL: We express our appreciation to the reviewers for providing valuable feedback that significantly enhanced the manuscript. We have thoroughly examined each comment and subsequently incorporated revisions into the manuscript. The modifications have been indicated by highlighting the text with the underlined words in red color in the Word document. Below, you will find comprehensive responses to all crucial points.

Reviewer 1: The authors present a comprehensive study on the enhancement of oxygen activity in LaFeO_3 perovskite oxides by creating subsurface La vacancies. Through a combination of experimental methods and theoretical calculations, it demonstrates that removing subsurface La cations significantly improves oxygen mobility and lowers the electron density of oxygen atoms. This leads to a 2.9-fold increase in methane conversion efficiency and high syngas selectivity (98%) in anaerobic oxidation, compared to stoichiometric LaFeO_3 . The study challenges conventional understanding and opens new avenues for designing redox catalysts, especially for reactions via the Mars–van Krevelen mechanism and chemical looping processes. I recommend it be published after the following comments are addressed:

[Reply] The authors appreciate for your positive comments.

Question 1: The authors should provide a more detailed explanation of how the subsurface La vacancies are formed, and how to control the vacancy concentration. This could include the specific conditions under which these vacancies are generated.

[Reply] The author sincerely appreciates your valuable suggestion. According to pioneering works, the perovskite oxide could accommodate A-site cation vacancies while maintaining a stable structure¹⁻³, which lays the foundation for generating subsurface A-site cation vacancies. Herein, we construct the subsurface La vacancies by two methods, including reducing the La/Fe ratio to below 1, e.g., $\text{La}_{0.97}\text{FeO}_3$ in this work, or conducting redox treatment as that of the chemical looping process (Figure 5 in the revised manuscript). The inherent mechanism and specific conditions to control the

concentration of subsurface La vacancies are described below:

1) In general, a lower La/Fe ratio than 1 in lanthanum ferrite would be helpful in increasing the percentage of Fe-O redox pair at oxide surface and constructing subsurface La vacancies (beneath surface Fe-O layer). This could improve the amount of active oxygen and the intrinsic reactivity of each oxygen, respectively. However, when La/Fe ratio is reduced to below 0.95, Fe₂O₃ impurities can be generated due to the collapse of perovskite matrix^{4,5}. In this regard, a La/Fe ratio between 0.95 and 1.0 would be optimal to modulate the concentration of subsurface La vacancies.

Figure R1. XRD patterns of fresh LaFeO₃, LaFeO₃ after CH₄ reduction and different regeneration time in CO₂ with corresponding magnified view in the range of 28-31° and 44.0–45.5°.

2) Redox treatment, such as chemical looping process, could promote the formation of subsurface La vacancies in stoichiometric LaFeO₃. As shown in Figure R1, when LaFeO₃ was reduced by CH₄ (marked as CH₄-Re), some of Fe cations were reduced to metallic Fe⁰ with production of La₂O₃ oxides ($\text{LaFeO}_3 + 1.5\text{CH}_4 \rightarrow 1.5\text{CO} + 3\text{H}_2 + \text{Fe}^0 + 0.5\text{La}_2\text{O}_3$). Subsequent oxidation by CO₂ could recover the lattice oxygen. After 4 min of oxidation (marked as CO₂-Ro-4min), it was found that no signals for Fe⁰ (or FeO_x) could be observed while that of La₂O₃ can be clearly detected, suggesting that the access of Fe cations back to LaFeO₃ perovskite matrix proceeds much faster than that of La

cations ($\text{LaFeO}_3 + x\text{Fe} + 1.5x\text{CO}_2 \rightarrow \text{LaFe}_{1+x}\text{O}_{3+1.5x} + 1.5x\text{CO}$; $\text{LaFe}_{1+x}\text{O}_{3+1.5x} + 0.5x\text{La}_2\text{O}_3 \rightarrow (1+x)\text{LaFeO}_3$). This is mainly due to the slow rate of A-site cation migration (cation size is much larger than that of B-site cation), which is a rate-controlling step for the formation of an ideal perovskite crystal structure with an A/B ratio of 1^{6, 7}. Therefore, the concentration of subsurface La vacancies can be increased by retarding the diffusion of La, such as reducing the regeneration time or lowering the temperature of CO₂ oxidation.

We have supplemented the related discussion in the revised manuscript (Page 13-15).

Question 2: They found an increase in particle size observed via SEM analysis post-redox cycling. How does this increase in particle size correlate with the catalytic activity, especially considering that larger particles often have a reduced surface area?

[Reply] This is a good question. As for stoichiometric LaFeO₃, the particle size (Figure S8 and Table S8 in the revised supplementary information) and the Fe/La ratio (Figure 5a in the revised manuscript) at surface greatly changed during the chemical looping reaction, which can exert great influence to the redox activity. A correlation between syngas productivity, specific surface area, and surface Fe percentage was compared, as shown in Figure R2. Corresponding parameters of La_{0.97}FeO₃ catalyst was also included for comparison. Although the percentage of surface Fe cations increased from 20 % (fresh LaFeO₃) to 58 % (LaFeO₃-3) after three redox cycles, the syngas yield notably declined from 4.7 mmol/g (LaFeO₃) to 1.8 mmol/g (LaFeO₃-3), which should be mainly induced by the notable decline of BET surface area (from 5.8 of LaFeO₃ to 1.3 m²/g of LaFeO₃-3). This result suggests that the specific surface area (or suitable particle size), which affects the number of active centers and diffusion of lattice oxygen from the bulk to surface, plays important role in modulating the reactivity of redox catalysts. When prolonging the tests to 20 cycles, the change of specific surface area (from 1.3 m²/g of LaFeO₃-3 to 1.8 m²/g of LaFeO₃-20) and surface Fe percentage is insignificant (from 58% of LaFeO₃-3 to 50% of LaFeO₃-20). However, the syngas yield notably increased from 1.8 mmol/g (LaFeO₃-3) to 4.6 mmol/g (LaFeO₃-20), which should be attributed to the formation of subsurface La

vacancies that greatly enhanced the intrinsic activity of surface oxygen. Moreover, the $\text{La}_{0.97}\text{FeO}_3$ sample with 3% La-deficiency that bearing 56 % surface Fe cations (Figure S17 in the revised supplementary information) and similar surface area of $2.2 \text{ m}^2/\text{g}$ gives a syngas yield of 5.2 mmol/g , which is 2.9 times higher than stoichiometric LaFeO_3 -3 sample. These results show that the oxygen activity can be greatly enhanced upon generating subsurface La-vacancies, substantially leading to the boosted performance for selective methane oxidation.

We have added the related description in the revised manuscript (Page 20-21).

Figure R2. Correlation between specific surface area, surface Fe percentage and the corresponding chemical looping methane conversion performance.

Question 3: The manuscript describes the migration of La cations out of the perovskite matrix during redox cycles, leading to surface reconstruction. It is unclear how this process specifically affects the stability.

[Reply] Thanks for the comment. We have conducted 250 cycles of the CH_4 - CO_2 redox cycle on LaFeO_3 oxygen carriers, observing that the redox performance (Figure 3i of the revised manuscript)

and particle size (Figure R3) of LaFeO_3 remain almost stable from the 20th cycle to 250th cycle. This suggests that migration of La out of the perovskite matrix, leading to both improved reactivity and sintering resistance, plays a key role in maintaining the redox stability in long-term chemical looping reactions.

As depicted in Figure 5a and elucidated by LEIS results, notable reduction in surface La content from 80% to 42% was observed during first three cycles of oxidation-reduction treatment (1st cycle to 3rd cycle), which indicates that the redox reaction can result in aggregation of the outmost La cations. Besides, the particle size significantly increases (Figure R3a and 3b), which leads to the syngas yield notably declined from 4.7 mmol/g (LaFeO_3) to 1.8 mmol/g (LaFeO_{3-3}) (Figure 3i in the revised manuscript). Upon extending the tests to 20 cycles, a slight increase of the surface La percentage from 42% to 50%, suggesting that continuous redox treatment induces the exsolution of additional La cations beneath the surface Fe in perovskite matrix, resulting in the generation of subsurface La vacancies (Figure 5a in the revised manuscript) and surface La enrichment (Figure 4 in the revised manuscript). The formation of subsurface La vacancies significantly reduces the barrier for CH_4 activation (Figure 6i in the revised manuscript), improving the syngas yield from 1.8 mmol/g in LaFeO_{3-3} to 4.6 mmol/g in LaFeO_{3-20} with a 2.5-fold increase (Figure 7 in the revised manuscript). Besides, from the 3rd cycle to the 20th cycle, the growth in particle size is inconspicuous (Figure R3b and 3c), suggesting that the surface reconstruction (surface La enrichment) could promote the sintering resistance. To double check above conclusion, the surface area and particle size of fresh and cycled La_xFeO_3 ($x = 1.03, 1, \text{ and } 0.97$) oxides were studied by BET and SEM. The surface area of fresh samples increases with improving La/Fe ratio (from 2.2 m^2/g for $\text{La}_{0.97}\text{FeO}_3$ to 9.4 m^2/g for $\text{La}_{1.03}\text{FeO}_3$) and corresponding SEM results also confirm that the particle size of fresh samples decreases with increasing the La content. After 20 redox cycles, it is noted that samples with higher La content maintain a smaller particle size, indicating that surface La enrichment is beneficial for inhibiting the sintering of oxygen carrier particles (Figure S8 in the revised supplementary information).

In summary, the migration of La cations out of the perovskite matrix during redox cycles leads to surface reconstruction that includes both the generation of subsurface La vacancies and surface La enrichment. Besides, with the extension of time, the surface structure of LaFeO₃ oxygen carrier reaches a dynamic equilibrium. Specifically, the subsurface La vacancies can promote the conversion of CH₄ to CO, and the surface La enrichment can inhibit sintering, which plays an important role in maintaining the long-term redox stability in chemical looping reactions.

We have added the related description in the revised manuscript (Page 11).

Figure R3. SEM images of (a) LaFeO₃, (b) LaFeO₃-3, (c) LaFeO₃-20, (d) LaFeO₃-250.

Question 4: The DFT calculations could be better correlated with the experimental results. The surface free energy of different possible surface structures (different terminated surfaces) should be calculated to confirm the model used in this work is reasonable and stable. At the same time, the effect of La vacancy at different locations, or different combinations of LaFeO₃, needs to be discussed.

[Reply] Thank you for the valuable suggestion. There are two different possible surface structures for LaFeO₃: the Fe-O terminated surface and the La-O terminated surface (Figure R4). In theory, the terminated surface structure with the lowest energy is considered to be the stable surface structure. As shown in Table R1, we found that the La-O terminated surface is the more stable surface. However,

compared to the Fe-O terminated surface, the La-O terminated surface is almost inert for the methane partial oxidation reaction⁸. Therefore, we choose the Fe-O terminated surface as our model, which has also been widely used for analyzing the methane conversion process over LaFeO₃ material⁹⁻¹¹, for DFT calculations.

Figure R4. Fe-O terminated surface for LaFeO₃ through (a) side view (c) top view, and La-O terminated surface for LaFeO₃ through (b) side view (d) top view.

Table R1. Total energy for different terminated surfaces.

Different terminated surface	Total energy (eV)
Fe-O terminated surface	-598.59
La-O terminated surface	-599.33

Inspired by your insightful comment, we further explored the structures with different La vacancies for LaFeO₃. As shown in Figure R5a and R5b, we discover that there are four different La

vacancy locations, *i.e.*, La1_{vac}, La2_{vac}, La3_{vac}, and La4_{vac}, because of the two different O sites in the LaFeO₃ structure, *i.e.*, top O (coordinated with La1 and La2) and bottom O (coordinated with La3 and La4). According to the structural symmetry, La1_{vac} and La3_{vac} are equivalent to La2_{vac} and La4_{vac}, respectively. As tabulated in Table R2, the total energy of LaFeO₃ with La1_{vac} and total energy with La3_{vac} are almost the same, indicating a single La vacancy is possibly generated in both La1_{vac} and La3_{vac} sites. However, the top O shows lower O vacancy formation energy than the bottom O in a single La vacancy structure for LaFeO₃ (Table R3), suggesting the top O is conducive to activating methane. It is worth noting that the structure with two La vacancies shows an identical trend with those of a single vacancy. Consequently, we choose the LaFeO₃ with La1_{vac} and La2_{vac} as the models for the following calculations (Figure R5c and 5d). Additionally, we present the results calculated based on the bottom O in Figure S14-15 and Table S5 for better reference.

We have added the related description in the revised manuscript (Page 15-17).

Figure R5. Different La vacancies for LaFeO₃ through (a) side view (b) top view. (c) LaFeO₃ with a single La vacancy (d) LaFeO₃ with double La vacancies.

Table R2. Total energy for different La vacancies.

Different La vacancies	Total energy (eV)
La1 _{vac}	-583.15
La3 _{vac}	-583.14
La1 _{vac} and La2 _{vac}	-567.09
La3 _{vac} and La4 _{vac}	-567.13

Table R3. Oxygen vacancy formation energy for different La vacancies.

Oxygen vacancy formation energy (eV)	Top O	Bottom O
0-La _{vac}	3.26	4.28
1-La _{vac}	1.37	1.82
2-La _{vac}	0.79	1.41

Question 5: In addition, to compare the activity of 0-La(vac), 1-La(vac), and 2-La(vac), the energy profile for the whole reaction on these surfaces should be provided.

[Reply] We fully agree with the reviewer. Considering your question 4, we have included the energy profile based on the top O for the entire reaction. As shown in Figure R6, the rate determining step for methane conversion is the activation of first C-H bond in CH₄ (TS1: CH₄*→CH₃*+H*) for LaFeO₃, corresponding to the previous reports¹². As for the catalyst with subsurface La vacancies, the energy barrier for TS1 was greatly reduced in an order of 1.88 eV (0-La_{vac}), 1.46 eV (1-La_{vac}), and 1.03 eV (2-La_{vac}), respectively (Table R4). Besides, it is found that the energy barriers for cleavage of other C-H bonds in *CH₃ or generation of syngas (H₂ and CO) were obviously reduced after generating subsurface La vacancies, which should result from the reduced electron intensity over lattice oxygen and the declined oxygen vacancy formation energy (Table S6 in the revised supplementary information). These results further highlight the promotion effect of subsurface La vacancies for improving the activity of oxygen, substantially enhancing the reactivity towards selective methane conversion.

Accordingly, motivated by your comment, we have completed the whole energy profile based on the top O for the three samples and added the relevant discussion in the revised manuscript (Page 17-19). Meanwhile, we also include the original energy profile based on the bottom O for TS1 in Figure S14-15 and Table S5 for your better reference.

Figure R6. Energy profile for methane partial oxidation reaction.

Table R4. Energy barrier for methane partial oxidation reaction.

Energy barrier (E _v)	0-La _{vac.}	1-La _{vac.}	2-La _{vac.}
TS1	1.88	1.46	1.03
TS2	1.84	1.39	0.99
TS3	1.67	1.22	0.77
TS4	1.82	1.23	0.75

Reviewer 2:

Question 1: Whether XPS was carried out in-situ or ex-situ? If it is ex-situ then there is possibility of oxidizing the oxides during exposure to air and it will mislead the real analysis.

[Reply] The authors appreciate your valuable suggestions. As for the XPS analysis, it was carried out under ex-situ conditions, since the high reaction temperature (900 °C) in chemical looping process is unable to be reached by in-situ XPS instrument. The samples subjected to XPS analysis were either fresh oxygen carriers (OCs) after calcination in air or the spent OCs after CO₂ oxidation. The Fe cations in these samples predominantly exist in the stable +3 valence state, rendering them relatively resistant to oxidation in air (Figure S7 and S10 in supplementary revised information). The fresh samples oxidized by air were directly tested by XPS analysis. Meanwhile, the samples after the chemical

looping reaction (oxidized by CO₂) were cooled down to room temperature in He atmosphere, which were then moved into a well-closed XPS transfer bin (Figure R7) in the glove box (Ar protected), thus protecting the samples from the contamination in air before XPS analysis. Therefore, the XPS analysis in this work would be helpful to gain the real chemical state of Fe and O ions in the OCs.

According to the XPS analysis, it is noted that, compared with fresh LaFeO₃ and La_{0.97}FeO₃ (calcined by air), Fe 2p spectra of LaFeO₃-20 and La_{0.97}FeO₃-20 samples that regenerated in CO₂ slightly shift to higher binding energy (Figure 4 of revised manuscript). This suggests the migration of La out of the perovskite matrix, induced by redox reactions, could slightly improve the valence state of Fe. To double check this conclusion, DFT analysis was conducted and showed that Bader charge (Table S6 of revised manuscript) of Fe was gradually increased from +1.67 (no La_{vac.}) to +1.69 (two La_{vac.} sites), which is in line with the XPS analysis and confirm that the XPS results are credible.

We greatly appreciate the kindly reminding and, for clarity, the details for XPS analysis were replenishment in the revised manuscript (Page 23).

Figure R7. Picture of the XPS transfer bin.

Question 2: Temperature programmed surface reaction (TPSR) and O₂-TPD is necessary to find out the oxygen vacancy generation and their correlation.

[Reply] Thanks for your kindly reminding. O₂-TPD, H₂-TPSR and CH₄-TPSR tests were conducted to study oxygen mobility and the generation of oxygen vacancies ([O_v]) in fresh OCs (La_{1.03}FeO₃, LaFeO₃, and La_{0.97}FeO₃) and the representative samples after redox reaction (LaFeO₃-3 and LaFeO₃-

20).

O₂-TPD experiments (Figure R8a) were performed by adsorption of O₂ at room temperature and then desorption in He, but no signals assigned to O₂ was detected. This should be induced by the low specific surface areas of these OCs, leading to trace amount of O₂ being adsorbed on surface oxygen vacancies. Besides, the lattice oxygen in LaFeO₃ is stable due to the high oxygen vacancy formation energy (>3.62 eV)¹³. In a word, the desorption of lattice oxygen to generate oxygen vacancy by simple thermal treatment in He is difficult.

Therefore, we want to study oxygen mobility and the generation of oxygen vacancies ([O_v]) via reduction of H₂ or CH₄ with higher oxygen extraction capacity. Firstly, the effect of oxygen mobility on the generation of oxygen vacancies (H₂ + O_L → H₂O + [O_v]) was evaluated using H₂-TPSR tests (Figure R8b). It is noted that the H₂ uptake for La_{1.03}FeO₃, LaFeO₃, and LaFeO₃-3cycle can be only observed at temperature above 700 °C. In contrast, two H₂ consumption peaks, locating at low temperature zone from 300 °C to 500 °C and high temperature above 700 °C, respectively, could be observed for La_{0.97}FeO₃ and LaFeO₃-20. Combined the structural analysis (XRD, HRTEM, and Raman), it is concluded that generation of subsurface La vacancies via redox treatments and reducing the La/Fe ratio can significantly enhance the activity and mobility of lattice oxygen, promoting the generation of oxygen vacancies. Additionally, CH₄-TPSR experiments were performed to investigate the generation of oxygen vacancies and evaluate the activity of lattice oxygen (O_L) towards methane oxidation (CH₄ + O_L → CO + 2H₂ + [O_v]). For fresh La_xFeO₃ oxides (x = 1.03, 1, and 0.97), a significant decrease in the emergence temperature of H₂ and CO signals was noted as the La content decreased, along with a gradual enhancement in the intensity of syngas signals (Figure R9a-c). Besides, for stoichiometric LaFeO₃, after redox treatment, we observed that methane conversion was greatly suppressed over LaFeO₃-3, as suggested by the reduced intensity for syngas production, while the reactivity of LaFeO₃-20 was almost identical with that of fresh LaFeO₃ (Figure R9b, 9d and 9e). These results suggest that generation of subsurface La vacancies via redox treatments and reducing the La/Fe

ratio can significantly promote the generation of oxygen vacancies and enhance the methane anaerobic oxidation.

Overall, these results show that generating subsurface La vacancies via redox treatments and reducing the La/Fe ratio are effective to enhance the lattice oxygen activity in lanthanum ferrite, which can greatly promote the formation of oxygen vacancies and selective methane oxidation. Accordingly, we have added the related discussion in the revised manuscript (Page 5).

Figure R8. (a) O₂-TPD profiles for La_xFeO₃ (x = 1.03, 1, and 0.97) of fresh and cycled oxides, and (b) H₂-TPR profiles for La_xFeO₃ (x = 1.03, 1, and 0.97) of fresh and cycled oxides.

Figure R9. CH₄-TPR profiles for (a) La_{1.03}FeO₃ (b) LaFeO₃ (c) La_{0.97}FeO₃ (d) LaFeO₃-3cycle (e) LaFeO₃-20cycle.

Question 3: Authors can also check whether generating subsurface La vacancy the methane activation

temperature is reduced or not? In my opinion this catalyst can activate methane at higher temperature.

[Reply] Thanks for your kindly reminding. Indeed, as the reviewer mentioned, the temperature for methane activation slightly increases after generating subsurface La vacancy. As shown in CH₄-TPSR results, the arise of CO signal, symbol for methane oxidation by lattice oxygen, can be observed at 718 °C for the fresh LaFeO₃ (Figure R9b). In contrast, generation of subsurface La vacancies after 20 redox treatments (LaFeO₃-20) induces the emergence of CO at higher temperature of 731 °C (Figure R9e). This is because the oxidation of methane by LaFeO₃ is influenced by two factors, including the activation rate of the C-H bond ($\text{CH}_4 \rightarrow *C + \text{H}_2$) and the migration rate of lattice oxygen ($*C + \text{O}_L \rightarrow \text{CO} + [\text{O}_v]$). The rate-controlling steps vary at different temperature stages. At relatively low temperature, the diffusion of lattice oxygen is difficult, which greatly limits the reactivity for methane oxidation. This also explains why most of the chemical looping reactions for methane conversion were conducted at temperature higher than 800 °C^{12, 14, 15}. In contrast, the rate of lattice oxygen migration significantly increases with rising reaction temperatures, making C-H bond activation the rate-controlling step for methane conversion¹⁶ (Figure 6i in the manuscript). As for LaFeO₃, redox treatment induces generation of subsurface La vacancies in LaFeO₃-20, which can promote the oxygen activity towards C-H bond activation (Figure 6i in the manuscript). Therefore, the formation of subsurface La vacancies greatly promotes the production of CO, rendering similar performance for LaFeO₃ and LaFeO₃-20 during the isothermal redox reaction at 900 °C. With the extension of time, the surface structure of LaFeO₃ oxygen carrier reaches a dynamic equilibrium, and the methane reaction rate remains basically stable in 250 cycles (Figure 3i in the manuscript).

Question 4: What about the time-on-stream effect? The authors should check the activity for at least 100 h time-on-stream.

[Reply] Thanks for your suggestion. We have conducted 108 hours (250 cycles) of CH₄-CO₂ redox reactions on LaFeO₃. As shown in Figure R10, LaFeO₃ still maintained high methane conversion rate (above 76%) and high CO selectivity (above 97%) during the long-term reaction, indicating that the

LaFeO₃ has high redox stability. XRD results (Figure R11a) show that the spent LaFeO₃ still maintains a stable perovskite structure after 108 h, and only a small amount of La₂O₃ is detected, which was caused by the outward diffusion of La cations from the subsurface sites. In addition, HRTEM images (Figure R11b) displayed well-resolved lattice fringes with homogeneous mapping of La, Fe, and O elements, which signifies the fine perovskite-type structure of LaFeO₃ after 108 h. After undergoing three cycles of oxidation-reduction treatment, the particle size of LaFeO₃ significantly increases, but as the number of cycles was extended to 20 or even 250 cycles, the growth of particle size slows down (Figure R3). We believe that surface La enrichment is beneficial for suppressing the sintering of oxygen carrier particles. This primarily results from the slow migration rate of A-site cations, leading to an enrichment of La cations on the surface, which significantly inhibits the sintering process. In summary, even after undergoing 108 h of cyclic stability test, LaFeO₃ still maintains good structural stability and stable redox performance. We have added the related description in the revised manuscript (Page 8-10).

Figure R10. The performance of CH₄ partial oxidation step for LaFeO₃ from 8 h to 108 h (20 cycle-250 cycle). Reaction conditions: 100 mg catalyst treated with 5% CH₄/He (15 mL/min) for 8 min during CH₄ partial oxidation step, 5% CO₂/He (15 mL/min) for 10 min during CO₂ regeneration step at 900 °C, and the reactor was purged with He for 4 min (20 ml/min) between partial oxidation and reoxidation step.

Figure R11. (a) XRD patterns of LaFeO₃ and LaFeO₃-108 h and the corresponding magnified view in the range of 29.0–31.0° (2 theta), (b) EDS maps and corresponding HRTEM images of LaFeO₃-250 cycle.

References

1. Liu, X., Mi, J., Shi, L., et al. In Situ Modulation of A-Site Vacancies in LaMnO_{3.15} Perovskite for Surface Lattice Oxygen Activation and Boosted Redox Reactions. *Angew. Chem. Int. Ed.* **60**, 26747-26754 (2021).
2. Gao, X., Fisher, C. A. J., Kimura, T., et al. Lithium Atom and A-Site Vacancy Distributions in Lanthanum Lithium Titanate. *Chem. Mater.* **25**, 1607-1614 (2013).
3. HÄRdtl, K. H. & Hennings, D. Distribution of A-Site and B-Site Vacancies in (Pb,La)(Ti,Zr)O₃ Ceramics. *J. Am. Ceram. Soc.* **55**, 230-231 (1972).
4. Zhu, Y., Zhou, W., Yu, J., et al. Enhancing Electrocatalytic Activity of Perovskite Oxides by Tuning Cation Deficiency for Oxygen Reduction and Evolution Reactions. *Chem. Mater.* **28**, 1691-1697 (2016).
5. V.C. Belessi, P.N. Trikalitis, A.K. Ladavos, T.V. Bakas & Pomonis, P. J. Structure and catalytic activity of La_{1-x}FeO₃ system (x=0.00, 0.05, 0.10, 0.15, 0.20, 0.25, 0.35) for the NO+CO reaction. *Appl. Catal. A* **177**, 53-68 (1999).
6. Buscaglia, M. T., Bassoli, M., Buscaglia, V. & Vormberg, R. Solid-State Synthesis of Nanocrystalline BaTiO₃: Reaction Kinetics and Powder Properties. *J. Am. Ceram. Soc.* **91**, 2862-2869 (2008).
7. Huang, C., Wang, X., Liu, X., Tian, M. & Zhang, T. Extensive analysis of the formation mechanism of BaSnO₃ by solid-state reaction between BaCO₃ and SnO₂. *J. Eur. Ceram. Soc.* **36**, 583-592 (2016).
8. Wu, M., Ma, S., Chen, S. & Xiang, W. Fe–O terminated LaFeO₃ perovskite oxide surface for low temperature toluene oxidation. *J. Clean. Prod.* **277**, 123224 (2020).

9. Polo-Garzon, F., Fung, V., Zhang, J., et al. CH₄ Activation over Perovskite Catalysts: True Density and Reactivity of Active Sites. *ACS Catal.* **12**, 11845-11853 (2022).
10. Chang, H., Bjørgum, E., Mihai, O., et al. Effects of oxygen mobility in La–Fe-based perovskites on the catalytic activity and selectivity of methane oxidation. *ACS Catal.* **10**, 3707-3719 (2020).
11. Kim, J., Kim, Y. J., Ferree, M., et al. In-situ exsolution of bimetallic CoFe nanoparticles on (La,Sr)FeO₃ perovskite: Its effect on electrocatalytic oxidative coupling of methane. *Appl. Catal. B* **321**, 122026 (2023).
12. Jie Yang, Erlend Bjørgum, Hui Chang, et al. On the ensemble requirement of fully selective chemical looping methane partial oxidation over La-Fe-based perovskites *Appl. Catal. B* **301**, 120788-120792 (2022).
13. Chang, W., Gao, Y., He, J., et al. Asymmetric coordination activated lattice oxygen in perovskite ferrites for selective anaerobic oxidation of methane. *J. Mater. Chem. A* **11**, 4651-4660 (2023).
14. Liao, X., Long, Y., Chen, Y., et al. Self-generated Ni nanoparticles/LaFeO₃ heterogeneous oxygen carrier for robust CO₂ utilization under a cyclic redox scheme. *Nano Energy* **89**, 106379 (2021).
15. Zhang, X. H., Pei, C. L., Chang, X., et al. FeO₆ octahedral distortion activates lattice oxygen in perovskite ferrite for methane partial oxidation coupled with CO₂ splitting. *J. Am. Chem. Soc.* **142**, 11540-11549 (2020).
16. Mihai, O., Chen, D. & Holmen, A. Chemical looping methane partial oxidation: The effect of the crystal size and O content of LaFeO₃. *J. Catal.* **293**, 175-185 (2012).

REVIEWERS' COMMENTS

Reviewer #1 (Remarks to the Author):

Thanks for addressing the comments. The authors have significantly improved the manuscript. I am happy to recommend it for publication.

Reviewer #2 (Remarks to the Author):

The authors have addressed almost all the concerns of the reviewers in a proper way, so the revised manuscript is recommended for publication